# Quantity, Duration, Adherence, and Reasons for Dietary Supplement Use among Adults: Results from NHANES 2011–2018

**DOI:** 10.3390/nu16121830

**Published:** 2024-06-11

**Authors:** Ligang Liu, Heqing Tao, Jinyu Xu, Lijun Liu, Milap C. Nahata

**Affiliations:** 1Institute of Therapeutic Innovations and Outcomes (ITIO), College of Pharmacy, The Ohio State University, Columbus, OH 43210, USA; liu.10645@osu.edu; 2Department of Gastroenterology, The First Affiliated Hospital, Guangzhou Medical University, Guangzhou 510182, China; tao_heqing@163.com; 3IT Research & Innovation, Nationwide Children’s Hospital, Columbus, OH 43210, USA; jinyu.xu@nationwidechildrens.org; 4Drake Institute for Teaching and Learning, The Ohio State University, Columbus, OH 43210, USA; liu.10667@osu.edu; 5College of Medicine, The Ohio State University, Columbus, OH 43210, USA

**Keywords:** dietary supplements, National Health and Nutrition Examination Survey, quantity, duration, reasons, adherence

## Abstract

Dietary supplement use is common among US adults. We aimed to investigate the quantity, duration, adherence, and reasons for supplement use in individuals who take supplements. Data from 2011 to 2018 from the National Health and Nutrition Examination Survey (NHANES) dataset were analyzed. Four cycles of data were combined to estimate these outcomes. Results are presented as overall group and by subgroups. All analyses were weighted to be nationally representative. The Taylor Series Linearization approach was used to generate variance estimates. A total of 12,529 participants were included. Over 70% of these individuals reported taking more than one unit of dietary supplements daily. Notably, approximately 40% had been taking supplements for more than five years and about 67% were highly adherent to at least one supplement. However, only 26.9% of these supplements were taken following a doctor’s recommendation. The primary reasons for dietary supplements intake included improving overall health (37.2%), maintaining health (34.7%), bone health (21.4%), and diet supplementation (20.3%). Our findings indicate that most participants proactively used multiple dietary supplements focused on self-managed health and prevention, with substantial dedication to long-term use and high adherence. Healthcare professionals should play a more active role in guiding such behaviors to optimize the health outcomes of dietary supplement users across the United States.

## 1. Introduction

The prevalence of dietary supplement (DS) use has been on the rise in the United States (US), reaching 56% of the population in 2018 [1]. Moreover, the demand for DSs had a significant surge during the COVID-19 pandemic [2,3]. Multivitamins and/or multimineral products stood out as the most used DSs [1,4]. Globally, the supplements market has had a substantial growth over the past decade and reached nearly $353 billion in 2019 [5]. 

Dietary supplements can improve health when taken appropriately [6]. Current public health policies do not advocate the use of supplements in optimally nourished populations [7]. Some commercially available products may potentially pose toxicity risks to consumers [8]. Vitamins or fish oil supplements are not recommended to reduce the risk of non-communicable diseases in people without nutritional deficiencies [9]. Dietary supplements did not show mortality benefits among US adults [10]. The available evidence is inadequate to evaluate the comparative benefits and risks of multivitamins for the prevention of cardiovascular disease or cancer [11].

Using National Health and Nutrition Examination Survey (NHANES) data from 2007–2010 [12], the main reasons for taking supplements were to improve or maintain overall health. However, in recent years, reasons driving dietary supplement use among supplement users have been unknown. Several demographic and socioeconomic characteristics are associated with supplement-taking behavior, including being female and those with older ages, higher educations, higher incomes, and healthy lifestyles [13,14,15,16,17,18,19]. Differences in health beliefs derived from culture, ethnicity, and other socioeconomic factors could lead to people using dietary supplements for different reasons [20,21]. The impacts of demographic and socioeconomic factors on these reasons have not been studied. About one-third of US adults may take one or more dietary supplements daily [22]. However, the quantity of supplements taken daily, the duration of supplement use, and adherence to dietary supplements’ intake remain inadequately established. 

The aims of this study were (1) to describe the quantity, duration, and adherence to dietary supplements; (2) to examine the reasons behind the use of dietary supplements; and (3) to investigate the association between demographic and socioeconomic characteristics on the top 10 reasons for dietary supplements’ use among individuals who used dietary supplements. Our hypothesis was that demographic and socioeconomic characteristics could impact the quantity, duration, adherence, and reasons for taking dietary supplements. 

## 2. Materials and Methods

The NHANES survey employed a complex, multistage sampling design to ensure the data collected were representative and inclusive of the diverse population within the US. Data were gathered through in-person interviews and physical examinations. Participants provided a wide array of information, including demographics, socioeconomic status, dietary patterns, medical history, dental health, physiological measurements, and laboratory results. To ensure the reliability and accuracy of the data, NHANES implemented stringent quality control procedures across all stages of data collection. These included calibration of equipment, training and re-training of field staff, and rigorous data review and validation processes before the dataset was made publicly available. 

Participants in the NHANES study provided informed consent and the cross-sectional survey design received approval from the National Center for Health Statistics Ethics Review Board. As this study utilized publicly available, anonymized data from NHANES, additional review by an institutional review board was not required.

The data utilized in this analysis were obtained from household interviews (N = 9756 from 2011 to 2012; N = 10,175 from 2013 to 2014; N = 9971 from 2015 to 2016; and N = 15,560 from 2017 to 2018). We excluded individuals who did not take any supplements (N = 27785), were younger than 20 years old (N = 4954), or were lactating or pregnant (N = 196) (Appendix A). All data were sourced from the demographics data and dietary data modules. 

### 2.1. Key Variables

The participants were initially asked whether they had taken any vitamins, minerals, herbals, or other dietary supplements in the past 30 days. For those who answered yes, hand cards were provided to record the reason(s) for taking each dietary supplement. Participants had the flexibility to select multiple reasons for taking a particular supplement. The commonly provided reasons from the hand cards are included in Appendix A. Furthermore, participants were asked whether the supplements were self-initiated or recommended by doctors. For each participant, the quantity of each DS was added to calculate the total quantity of dietary supplements taken daily. The unit of measurement for dietary supplements could be caplets, capsules, tablets, etc. The summed quantities of distinct supplements taken were categorized as follows: ≤1, 1.1 to 4.9, 5 to 10, and >10. Participants were also asked to specify the duration of use for each supplement, measured in days. The duration of supplement use was divided into less than a year (365 days), 1 to 5 years (365 to 1825 days), 5 to 10 years (1836 to 3650 days), and greater than 10 years (3650 days). Adherence to supplement intake was assessed by determining the number of days any supplement was taken within the intended past 30 days. The highest number of days was used to represent the overall intake if a participant took multiple supplements. Adherence was classified as low (<50% or <15 days), moderate (50% to 79% or 16 to 23 days), and high (80% or ≥24 days) during the 30-day period. 

The demographic module provided age, gender, race and ethnicity, education level, and the ratio of family income to poverty. To analyze the ratio of family income to poverty, three categories were established: less than 130%, 130% to 350%, and more than 350% of the national poverty level. Race and ethnicity were classified as Hispanic, non-Hispanic White, non-Hispanic black, non-Hispanic Asian, and other races. Education level was categorized as less than high school, high school, some college, and college graduate or above. Age groups were defined as 20–39 years, 40–64 years, 65–79 years, and 80 years and above.

### 2.2. Statistical Analyses

All statistical analyses were performed using R (4.3.1). Four continuous cycles (2011 to 2018) were combined to investigate the quantity, duration, adherence, and main reasons for taking supplements. The Taylor Series Linearization approach was used to generate variance estimates (percentages with standard errors (SEs) in accordance with NHANES guidelines. To investigate the associations between specific reasons for taking supplements and demographic and socioeconomic characteristics, including sex, age, race/ethnicity, education level, and income level, the unadjusted and adjusted odds ratios (ORs) with 95% confidence intervals (CIs) were calculated using univariate and multivariate logistic regression models. Initially, we conducted univariate logistic regression analyses to assess the individual effects of each variable on the reasons for dietary supplement intake. For the multivariate regression analysis, we included all five variables, adjusting each one independently in the model. This approach allowed us to obtain adjusted odds ratios that reflect the influence of each variable while controlling for the effects of the others. A fixed entry method was used for variable inclusion. The median values and interquartile ranges (IQRs) for supplement usage metrics (quantity, duration of use, and adherence in the past 30 days) across different demographic groups were calculated because the distributions were not symmetric. Mann–Whitney U tests were used to determine the statistically significant differences between groups. The common reasons for using supplements, whether by personal choice or doctor’s recommendation, were compared and analyzed using Chi-square tests. Additionally, a comparison between the common reasons reported by women and men within each group was conducted. After a preliminary assessment of the dataset, we observed that the incidence of missing values was relatively low. Consequently, we employed a complete case analysis approach. A 2-sided *p*  <  0.05 was used to establish statistical significance. Weighting was assigned to all participants to account for their unequal sampling probability and non-response. 

## 3. Results

A total of 12,529 individuals were included in this study. Most of them were female (56.1%), middle-aged (43.8%), white (42.7%), held a college degree or higher (61.9%), and earned an income above 130% of the poverty level (73%). The baseline demographics, detailed in Table 1, provide a snapshot of the socio-economic and educational background of the cohort.

### 3.1. Quantity, Duration, and Adherence of Supplements 

Overall, over 70% of participants took more than one unit of supplement daily. Individuals who were female, White or Hispanic, with increased age, of a higher education level, and of a higher income were likely to take a higher quantity of supplements, and 67% of the participants demonstrated high adherence to at least one dietary supplement. With increased age and higher income, more individuals had higher adherence. Compared with males, females also had higher adherence. More than 20% of the population had been using dietary supplements for more than 10 years. People who were white, older adults, and with a higher education level and a higher income took supplements for a longer duration (Figure 1, Figure 2 and Figure 3 and Table 2).

### 3.2. Reasons for Taking Supplements

Participants could choose more than one reason for taking dietary supplements. The top 10 reasons were for overall health improvement (37.2%), health maintenance (34.7%), bone health (21.4%), diet supplementation (20.3%), more energy (18.1%), disease prevention (16.7%), immunity enhancement (16.3%), and for heart (14.2%), joint (10.5%), and skin, hair, and nail health (9.4%) (Table 3 and Figure 4). Other reasons for taking supplements by different subgroups are reported in Appendix A. 

#### 3.2.1. Reasons for Taking Supplements by Own Initiative

More than two-thirds (73.1%) of the supplements were taken by participants based on their own discretion. The most prevalent motivations for supplement intake were to improve overall health (46.6%), maintain health (43.4%), promote bone health (22.8%), and supplement diet (23.5%). Furthermore, the use of supplements for more energy, bone health, and healthy skin, hair, and nails were more commonly observed among women compared to men. 

#### 3.2.2. Recommended Reasons from Health Professionals for Supplement Use

About one-fourth (26.9%) of the supplements were taken from the advice of health professionals. Healthcare providers commonly suggested supplements for health improvement (37.7%), health maintenance (36.1%), bone health (35.4%), diet supplementation (24.7%), and heart health (22.2%) (Table 4). Healthcare providers were more likely to encourage use of supplements for bone health, healthy skin, hair, and nail, and more energy for females. Conversely, the reasons of maintaining health, preventing disease, and promoting heart health were more prevalent among males than females. Supplements for heart and bone health were more commonly recommended by healthcare professionals than being self-initiated by individuals. Reasons of improving overall health, preventing disease, maintaining health, boosting the immune system, supporting skin, hair, and nails, and enhancing energy levels were more frequently initiated by participants themselves rather than being suggested by doctors.

#### 3.2.3. Effect of Sex on Reasons for Supplement Use 

Supplement use for bone health (OR: 2.81, 95% CI: 2.4–3.29), healthy skin, hair, and nails (OR: 3.61, 95% CI: 2.99–4.37), and more energy (OR: 1.19, 95% CI: 1.06–1.34) were significantly more frequent among females than males. In the multivariable regression model, females also were more likely to take supplements for diet supplementation (OR: 1.15, 95% CI: 1.01–1.3) and joint health (OR: 1.2, 95% CI: 1.01–1.44). More males took supplements to improve overall health (OR: 0.91, 95% CI: 0.83–0.99) versus females (Figure 4A and Appendix A) 

#### 3.2.4. Effect of Age on Reasons for Supplement Use 

There was a positive trend for increased use of dietary supplements for heart, joint, and bone health as age increased. However, these increases in people who were greater than 80 years of age were not as significant as those in the 65–79 years age group (Figure 1). We also observed a negative trend of fewer people using supplements for energy, immune boosting, and skin, hair, and nail health with increasing age (Figure 4). There was also a trend of decreased supplement use for muscle health, relaxation, sleep, stress, and weight loss as age increased (Appendix A).

#### 3.2.5. Effect of Race/Ethnicity on Reasons for Supplement Use 

We observed notable differences in some reasons for supplement use among racial groups. In comparison to individuals of White ethnicity, Hispanics tended to take supplements more frequently for healthy skin, hair, and nails (OR: 1.35, 95% CI: 1.12–1.63) and for higher energy levels (OR: 1.68, 95% CI: 1.47–1.92). In contrast, they were less inclined to take supplements for improving overall health, maintaining health, supplementing diet, preventing disease, and heart health. Interestingly, different results were noted in the multivariable regression analysis (Figure 1C and Appendix A). 

Likewise, in comparison to White adults, Black individuals tended to use supplements more often for healthy skin, hair, and nails (OR: 1.53, 95% CI: 1.21–1.92) and more energy (OR: 1.91, 95% CI: 1.65–2.21). Conversely, they took supplements less frequently for overall health maintenance, health improvement, and heart health. Fewer Asian adults took supplements for overall health, disease prevention, immunity enhancement, and increased energy compared to White adults (Figure 1C and Appendix A).

#### 3.2.6. Effect of Income and Education Level on Reasons for Supplement Use 

Individuals with a college degree or higher education tended to take dietary supplements more frequently to maintain or improve health, supplement diet, prevent health problems, prevent colds, and support joint, skin, nail, and hair health compared to those with less education than a high school degree. However, the reasons for more energy did not exhibit significant change between different educational groups. These trends are represented in Figure 1D,E and Appendix A.

## 4. Discussion

This study found that the top 10 reasons for taking dietary supplements were for improving overall health, maintaining health, bone health, diet supplementation, energy enhancement, preventing disease, immunity boost, heart health, joint health, and healthy skin, hair, and nails among US adults during 2011 to 2018 period. Most reasons varied based on age, gender, race/ethnicity, education level, and income level. These demographic factors played a significant role in shaping the reasons behind supplement intake. Most participants reported taking more than one unit of supplements daily for more than one year and demonstrated high adherence to supplements. Women, older individuals, and adults of White ethnicity tended to take a higher quantity of dietary supplements and were more likely to be highly adherent to supplements. Moreover, individuals of older age, White ethnicity, higher educational attainment, and increased income levels were more inclined to take supplements for extended durations.

Our findings are consistent with a previous study indicating that the primary reasons for supplement use revolved around overall health and bone health [12]. This suggests that the motivations behind supplement intake have remained relatively stable over the years. US adults were more inclined to take supplements to maintain or improve overall health, prevent diseases, boost their immune system, and enhance energy levels. This behavior reflects a desire for self-care and harm reduction with supplement use [23]. In our study, it was evident that only about one-fourth of the supplements were recommended by doctors. Among these recommendations, they commonly suggested supplements for health improvement and maintenance, bone health, diet supplementation, and heart health. Previous research reported the most common reasons of taking supplements recommended by a health professional were to maintain overall health and wellness, to prevent diseases, and to supplement diet [24,25]. Another study found that over 50% of health professionals recommended dietary supplements, primarily for heart health, skin, hair, and nail benefits and bone and joint health [26]. Dietitians also recommended dietary supplements for bone health, nutrient gaps, and overall health and wellness [27]. However, it has been suggested that dietitians should not recommend dietary supplements to prevent chronic diseases [28]. 

Our study revealed that more females than males took dietary supplements for bone health, and healthy skin, hair, and nails, both based on a health professional recommendation and their personal choice. Previous studies have also reported that calcium and biotin intake tended to be higher among females than males [29,30,31]. 

As individuals aged, certain motivations for supplement use tended to change. Older adults were less prone to experiencing sleep disorders and more likely to undergo weight loss compared to younger adults [32,33]. Previous research indicated a lower prevalence of melatonin and skin, hair, and nail health supplement use among older adults than in younger individuals [31,34]. Our observations also align with previous studies showing a declining trend in the prevalence of motivations such as more energy, healthy skin, hair, and nails, relaxation, sleep, stress, and weight loss as individuals became older. With the aging process, individuals are more likely to encounter disorders related to organs such as bones, joints, eyes, and other specific health conditions [35,36,37,38]. Overall, we observed a positive trend in motivations related to organ-specific reasons, including bone, joint, heart, bowel, kidney, bladder, and eye health, as individuals aged.

The use of multiple dietary supplements increases with advancing age and about a quarter of adults aged 60 years and over have taken four or more dietary supplements [22]. In a previous study, 76% of participants expressed the belief that supplements held important equivalence to prescriptions [24]. In our study, we observed that approximately 20% of participants consumed more than five units of supplements daily. We also discovered that approximately two-thirds of the participants took at least one supplement for 24 days during a 30-day period, indicating high adherence to dietary supplements. Individuals who were women, White, older, and with higher education and income levels demonstrated an increased prevalence of taking higher quantities of supplements and high adherence. However, the increased pill burden from dietary supplements could potentially lead to nonadherence to prescribed medications. Research has suggested that reducing this pill burden could enhance medication adherence [39]. 

The higher use of supplements for the health of bone, skin, hair, and nails by females could reflect greater health awareness among women. These trends suggest that interventions aimed at educating women about the evidence-based benefits and potential risks of supplements could be beneficial. The varied reasons for supplement use among different ethnic groups suggest cultural beliefs and socioeconomic factors can influence health behaviors. Tailoring public health messaging to address these cultural nuances and improve the understanding of supplements and risks to health can help mitigate misuse. Health education programs that are culturally sensitive and accessible might be more effective in conveying these messages.

Understanding these demographic patterns can also aid regulatory bodies in monitoring and regulating the supplement industry more effectively, ensuring that the marketed benefits are scientifically supported. Educating healthcare providers about these demographic trends in supplement use can prepare them to better manage discussions about supplement use with their patients, ensuring that they provide informed, personalized advice based on the latest research and patient demographics.

It is crucial for clinicians to be aware of the dietary supplements being taken by their patients and actively assist them in making informed decisions regarding their appropriate use [40]. However, only 25% of survey respondents reported having interactions about dietary supplements [41]. Many patients did not disclose supplement use if their provider did not ask [42]. Therefore, providing detailed information about dietary supplements to patients is important [43]. The majority of dietitians expressed a keen interest in continuing their education on dietary supplements [27]. To improve patient outcomes, training and educational programs should be extended to all healthcare professionals to enhance their knowledge about dietary supplements.

This study provides valuable insights into the real-world patterns of supplement use, highlighting the necessity for healthcare providers, including physicians and other healthcare practitioners, to discuss supplement use during routine consultations and offer evidence-based guidance. By understanding the patterns of supplement use, healthcare providers can deliver more tailored advice and public health officials and policy makers can utilize these insights to formulate public health messages and regulatory actions that ensure the safety and efficacy of dietary supplements on the market. Additionally, the data serve as a crucial resource for educators and researchers. Educators in nutrition can integrate these findings into their curriculum to enhance dietetics education. For researchers, the study raises important questions about the long-term health impacts of supplement use, identifying areas for further investigation.

This study has several limitations. First, the NHANES database consists of serial cross-sectional survey data from noninstitutionalized civilian adults in the US. Therefore, the generalizability of our findings might be limited to similar populations and settings. This limitation also arises from the self-reported nature of the dietary and demographic data within the NHANES database, introducing the possibility of recall bias. We did not include information regarding tobacco and alcohol use in this study due to the substantial amount of missing data in the original dataset. Presence of disease and body composition were not included in our analysis because they were not within the scope of our study. Future research should explore the effect of underlying disease and body composition on patterns of dietary supplement use. Additionally, our study did not delve into the prevalence of dietary supplement use and specific dietary supplements used because these have been extensively explored in other published studies. We did not explore the long-term health outcomes of dietary supplement use. Future studies could potentially address this gap to provide an understanding of the long-term implications of dietary supplement consumption.

## 5. Conclusions

Supplement-taking behavior was driven by different reasons and the reasons were heterogeneous by population subgroups. Most participants used multiple dietary supplements with the primary intention of enhancing or maintaining health with high adherence and long-term consumption. A relatively small percentage of supplements were taken following a physician’s guidance. The findings of this study underscore the practical necessity for healthcare providers to routinely inquire about dietary supplement use during patient consultations. This practice can enable the provision of evidence-based guidance on the safe and effective use of supplements. Healthcare providers should proactively discuss dietary supplement use with patients to ensure safe and effective use. By understanding the findings of this study, public health officials can develop targeted educational campaigns promoting informed choices regarding the safe and effective use of supplements.

## Figures and Tables

**Figure 1 nutrients-16-01830-f001:**
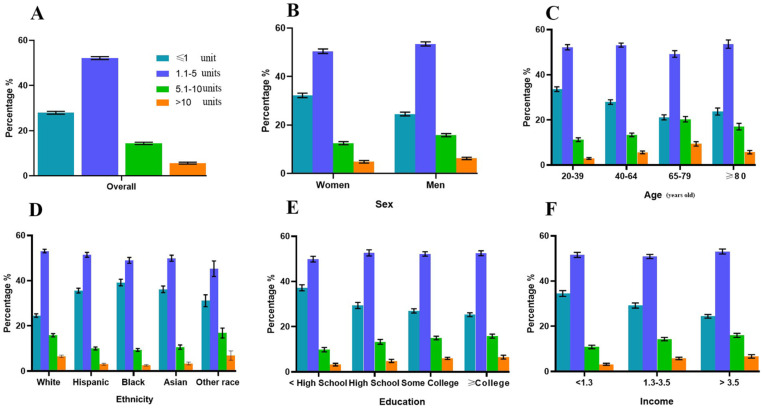
Quantity of daily dietary supplements taken by adults (≥20 years) who reported supplement intake overall (**A**) and by sex (**B**), age (**C**), race/ethnicity (**D**), education level (**E**), and income level (**F**), United States, 2011–2018. Data are presented as percentages (standard errors [error bars]) for dietary supplement users.

**Figure 2 nutrients-16-01830-f002:**
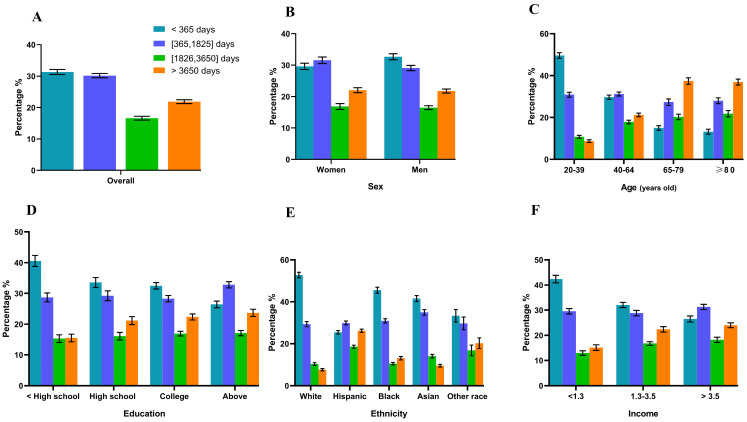
Days of dietary supplements used by adults (≥20 years) who reported supplement intake overall (**A**) and by sex (**B**), age (**C**), education level (**D**), race/ethnicity (**E**), and income level (**F**), United States, 2011–2018. Data are presented as percentages (standard errors [error bars]) for dietary supplement users.

**Figure 3 nutrients-16-01830-f003:**
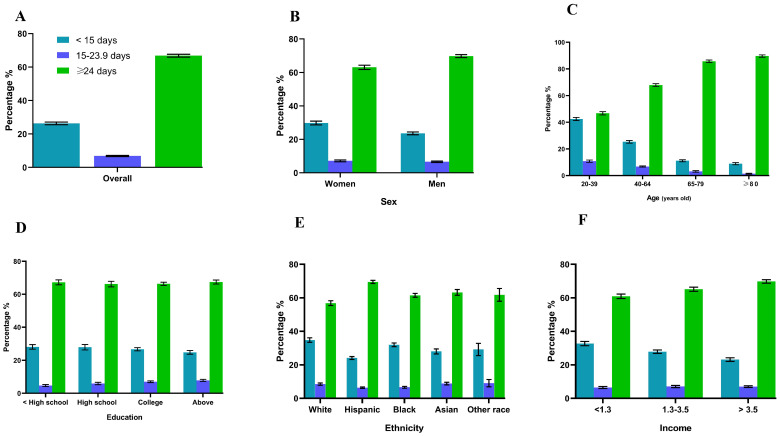
The number of days that dietary supplements were used in the past 30 days by adults (≥20 years) who reported supplement intake overall (**A**) and by sex (**B**), age (**C**), education level (**D**), race/ethnicity (**E**), and income level (**F**), United States, 2011–2018. Data are presented as percentages (standard errors [error bars]) for dietary supplement users.

**Figure 4 nutrients-16-01830-f004:**
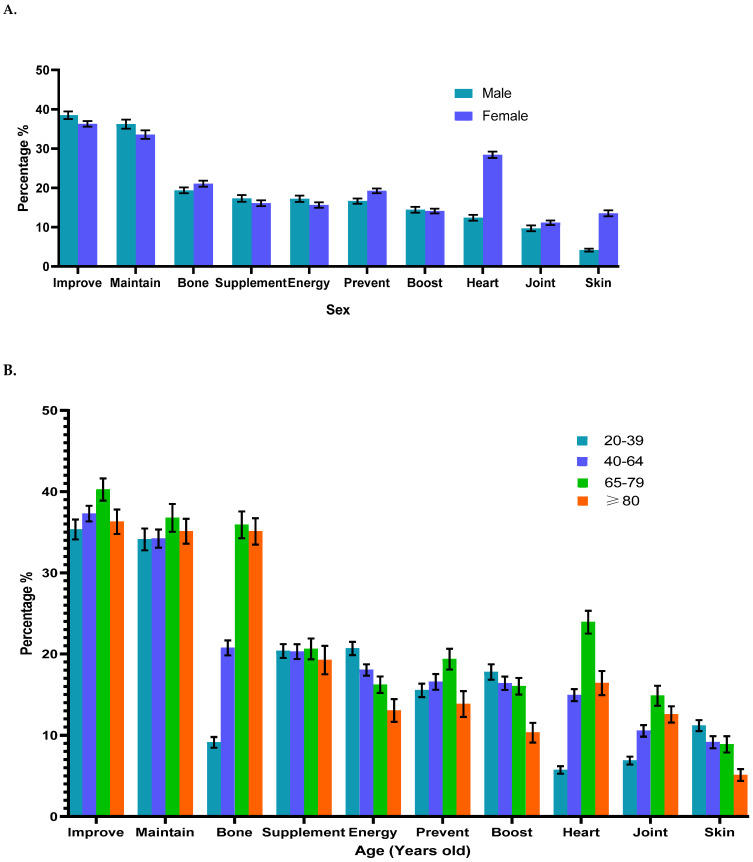
Prevalence of reported top 10 reasons for dietary supplement use among adults (≥20 years) by sex (**A**), age (**B**), race/ethnicity (**C**), education level (**D**), and income level **(E**) in the United States, 2011 to 2018. Improve indicates “improve overall health”; prevent, “prevent health problems”; maintain, “maintain health”; supplement, “supplement the diet”; boost, “boost the immune system”; energy, “increase energy”; heart, “for heart health or to lower cholesterol”; joint, “for healthy joints or arthritis”; skin, “for healthy skin, hair, and nails”; and bone, “for bone health”. Data are presented as percentages (standard errors [error bars]) for dietary supplement users.

**Table 1 nutrients-16-01830-t001:** Baseline characteristics among U.S. adults (≥20 years) who took supplements, NHANES 2011–2018.

	N (%)				
	2011–2012 (n = 2947)	2013–2014 (n = 3059)	2015–2016 (n = 3179)	2017–2018 (n = 3344)	Overall (n = 12,529)
Sex					
Male	1295 (43.9)	1305 (42.7)	1402 (44.1)	1496 (44.7)	5498 (43.9)
Female	1652 (56.1)	1754 (57.3)	1777 (55.9)	1848 (55.3)	7031 (56.1)
Age range, y					
20–39	808 (27.4)	760 (24.8)	792 (24.9)	727 (21.7)	3087 (24.6)
40–64	1273 (43.2)	1377 (45.0)	1383 (43.5)	1460 (43.7)	5493 (43.8)
65–79	604 (20.5)	646 (21.1)	717 (22.6)	814 (24.3)	2781 (22.2)
≥80	262 (8.9)	276 (9.0)	287 (9.0	343 (10.3)	1168 (9.3)
Race/ethnicity					
Hispanic	458 (15.5)	572 (18.7)	852 (26.8)	665 (19.9)	2547 (20.3)
Non-Hispanic White	1296 (44.0)	1512 (49.4)	1217 (38.3)	1317 (39.4)	5342 (42.6)
Non-Hispanic Black	687 (23.3)	512 (16.7)	601 (18.9)	708 (21.2)	2508 (20.0)
Non-Hispanic Asian	415 (14.1)	381 (12.5)	393 (12.4)	477 (14.3)	1666 (13.3)
Other Races	91 (3.1)	82 (2.7)	116 (3.6)	177 (5.3)	466 (3.7)
Education level					
<High school	550 (18.7)	515 (16.8)	598 (18.8)	533 (15.9)	2196 (17.5)
High school	562 (19.2)	648 (21.2)	631 (19.8)	739 (22.1)	2583 (20.6)
Some college	916 (31.1)	966 (31.6)	994 (31.3)	1144 (34.2)	4016 (32.1)
≥College	913 (31.0)	928 (30.3)	953 (30.1)	928 (27.8)	3731 (29.7)
Poverty income ratio					
<130%	828 (30.6)	793 (27.9)	735 (26.1)	683 (23.6)	3039 (27.0)
130–350%	889 (32.8)	984 (34.7)	1140 (40.5)	1197 (41.3)	4210 (37.4)
>350%	990 (36.6)	1062 (37.4)	938 (33.3)	1016 (35.1)	4006 (35.6)

**Table 2 nutrients-16-01830-t002:** Supplement intake patterns on the quantity of dietary supplements taken daily, the number of days any supplements were taken in the past 30 days, and the total days any supplement had been used, stratified by various demographic characteristics.

	Quantity of Supplement Taken Daily		Days Any Supplement Taken in the Past 30 Days		Days Any Supplement Has Been Taken	
	Median [IQR]	*p*-Value	Median [IQR]	*p*-Value	Median [IQR]	*p*-Value
Sex						
Men	2.00 [1.00, 4.00]	Reference	30 [15, 30]	Reference	1095 [365, 3650]	Reference
Women	2.00 [1.00, 4.25]	<0.001	30 [16.25, 30]	<0.001	1095 [274, 3650]	0.453
Race/ethnicity						
Non-Hispanic White	2.00 [1.00, 4.00]	Reference	30 [20, 30]	Reference	1825 [547, 4745]	Reference
Hispanic	3.00 [1.00, 5.00]	<0.001	30 [15, 30]	<0.001	547 [91, 1825]	<0.001
Non-Hispanic Black	2.00 [1.00, 3.00]	0.30473	30 [15, 30]	<0.001	730 [182, 2190]	<0.001
Non-Hispanic Asian	2.00 [1.00, 4.00]	0.73291	30 [15, 30]	0.0405	730 [182, 1825]	<0.001
Other Races	2.00 [1.00, 5.00]	0.00211	30 [12, 30]	0.0372	1095 [365, 3650]	0.0316
Age, y						
20–39	2.00 [1.00, 4.00]	Reference	20 [7, 30]	Reference	365 [91, 1460]	Reference
40–64	2.00 [1.00, 4.00]	<0.001	30 [15, 30]	<0.001	1095 [365, 3650]	<0.001
65–79	3.00 [1.00, 5.00]	<0.001	30 [30, 30]	<0.001	1825 [730, 5475]	<0.001
≥80	3.00 [1.00, 5.00]	<0.001	30 [30, 30]	<0.001	3650 [730, 7300]	<0.001
Education level						
Less than high school	2.00 [1.00, 3.00]	Reference	30 [15, 30]	Reference	730 [122, 2190]	Reference
High school	2.00 [1.00, 4.00]	<0.001	30 [15, 30]	0.684	1095 [274, 3650]	<0.001
Some college	2.00 [1.00, 4.00]	<0.001	30 [15, 30]	0.975	1095 [365, 3650]	<0.001
≥College	2.38 [1.00, 5.00]	<0.001	30 [15, 30]	0.335	1460 [365, 3650]	<0.001
Poverty income ratio						
<130%	2.00 [1.00, 3.00]	Reference	30 [14, 30]	Reference	365 [91, 1825]	Reference
130–350%	2.00 [1.00, 4.00]	<0.001	30 [15, 30]	0.00125	730 [182, 2555]	<0.001
>350%	2.00 [1.00, 4.00]	<0.001	30 [15, 30]	<0.001	1095 [365, 3650]	<0.001

Abbreviation: interquartile range (IQR).

**Table 3 nutrients-16-01830-t003:** Prevalence of reported top 10 reasons for dietary supplement use among adults (≥20 years) by sex, age, race/ethnicity, education level, and income level in the United States in 2011–2018. Data are presented as percentages (standard errors) for dietary supplement users.

				% (SE)						
Reason	Improve Overall Health	Maintain Health	Bone Health	Supplement the Diet	Enhanced Energy	Prevent Health Problems	Boost Immune System	Heart Health	Joint Health	Skin, Hair, and Nail
Overall	37.3 (0.7)	34.7 (0.8)	21.4 (0.6)	20.3 (0.6)	18.1 (0.5)	16.7 (0.7)	16.3 (0.6)	14.2 (0.5)	10.5 (0.5)	9.4 (0.5)
Sex										
Men	38.5 (1.0)	36.2 (1.2)	12.4 (0.8)	19.4 (0.7)	16.6 (0.7)	17.3 (0.9)	17.2 (0.8)	14.4 (0.7)	9.7 (0.8)	4.2 (0.4)
Women	36.3 (0.7)	33.6 (1.1)	28.4 (0.8)	21.1 (0.8)	19.2 (0.6)	16.1 (0.7)	15.6 (0.7)	14.1 (0.6)	13.5 (0.8)	11.1 (0.6)
Age range, y										
20–39	35.3 (1.2)	34.1 (1.4)	9.1 (0.7)	20.4 (0.9)	20.7 (0.8)	15.5 (0.8)	17.8 (1.0)	5.7 (0.5)	6.9 (0.5)	11.2 (0.7)
40–64	37.3 (1.0)	34.2 (1.1)	20.8 (0.9)	20.3 (0.9)	18.0 (0.7)	16.6 (1.0)	16.4 (0.8)	14.9 (0.7)	10.5 (0.7)	9.2 (0.7)
65–79	40.3 (1.4)	36.8 (1.7)	35.9 (1.7)	20.6 (1.3)	16.2 (1.0)	19.4 (1.3)	16.0 (1.0)	23.9 (1.4)	14.9 (1.2)	8.9 (1.0)
≥80	36.3 (1.5)	35.1 (1.5)	35.1 (1.6)	19.3 (1.8)	13.0 (1.4)	13.8 (1.6)	10.3 (1.2)	16.4 (1.5)	12.6 (1)	5.1 (0.7)
Race/ethnicity										
Non-Hispanic White	39.2 (0.8)	36.0 (1.0)	21.7 (0.7)	21.1 (0.7)	16.2 (0.6)	17.5 (0.9)	16.3 (0.8)	15 (0.7)	10.8 (0.7)	8.7 (0.6)
Non-Hispanic Black	34.6 (1.0)	31.2 (1.3)	19.1 (1.1)	18.4 (1.2)	27.0 (1.1)	15.1 (1)	17.5 (0.8)	12.1 (0.7)	9.2 (0.7)	12.6 (0.9)
Hispanic	29.4 (1.2)	28.3 (1.1)	20.3 (0.9)	16.4 (0.9)	24.6 (1.0)	13.6 (1.0)	15.0 (1)	10.5 (0.9)	8.8 (0.6)	11.4 (0.8)
Asian	35.3 (1.5)	37.4 (1.0)	22.9 (1.4)	20.9 (1.3)	12.7 (0.9)	13.4 (0.9)	12.5 (0.9)	14 (0.9)	10.4 (0.9)	10.1 (0.9)
Other races	33.1 (2.8)	35.1 (3.5)	22.8 (3.4)	21.7 (2.6)	20.8 (2.5)	17.6 (2.6)	24.3 (3.5)	16.5 (2.3)	12.2 (1.8)	9.2 (1.6)
Education										
<High school	25.3 (1.4)	25.4 (1.6)	22.1 (1.1)	14.4 (0.9)	20.3 (1.2)	11.4 (1.2)	11.5 (1.3)	11.4 (1.3)	8.5 (0.7)	6.6 (0.6)
High school	36.2 (1.3)	30.0 (1.4)	22.3 (1.3)	17.4 (1)	19.4 (1.0)	13.4 (0.9)	14.0 (1.3)	15.3 (1.2)	10.1 (1.0)	9.2 (1.2)
Some college	39.1 (1.2)	33.8 (1.2)	19.7 (0.8)	20.1 (1)	19.1 (0.8)	16.7 (0.9)	17.5 (0.9)	14.3 (0.7)	11.4 (0.8)	10.5 (0.7)
≥College	39.6 (1.1)	40.9 (1.1)	22.2 (1.0)	23.8 (0.9)	15.9 (0.9)	20 (1.0)	17.9 (1)	14.4 (0.8)	10.5 (0.8)	9.3 (0.6)
Poverty income ratio										
<130%	31.2 (1.4)	26.2 (1.4)	19.6 (1.0)	17.3 (1)	20.5 (1.1)	12.4 (1.0)	14.2 (0.9)	11.5 (1)	8.7 (0.8)	10.3 (0.8)
130–350%	38 (1.2)	33.5 (0.9)	21.1 (0.8)	18.7 (0.9)	20.4 (0.9)	16.9 (1)	18 (1.0)	15.3 (0.7)	10 (0.6)	9.2 (0.8)
>350%	39.8 (1.0)	38.8 (1.1)	22 (1.1)	23 (1)	16 (0.8)	18.6 (0.9)	16.9 (0.9)	15 (0.8)	11.8 (0.9)	9.1 (0.7)

**Table 4 nutrients-16-01830-t004:** Prevalence of reported reasons for dietary supplement use among adults (≥20 years) by self-initiated decision or doctors’ recommendation in the United States, 2011–2018.

	% (SE)					
	Self-Initiated Decision			Doctors’ Recommendation		
	Overall	Male	Female	Overall	Male	Female
To prevent health problems	20.7 (0.8)	21.1 (1)	20.3 (1)	18.1 (1) ^a^	20.4 (1.6)	16.9 (1.1) ^c^
To improve overall health	46.6 (0.8)	47.8 (1.1)	45.5 (1)	37.7 (0.9) ^a^	39.9 (1.8)	36.6 (1.1)
To supplement diet	23.5 (0.8)	22.6 (1)	24.2 (1)	24.8 (1)	23.9 (1.5)	25.2 (1.2)
To maintain health	43.4 (0.8)	45.3 (1.4)	41.8 (1.1)	36.1 (1.4) ^a^	39.5 (1.7)	34.4 (1.7) ^c^
boost immune system	21.7 (0.8)	22.3 (1)	21.1 (1)	14.6 (0.6) ^a^	16.2 (1.3)	13.8 (0.8)
For heart health, cholesterol	16.8 (0.6)	16.7 (0.9)	16.9 (0.8)	22.2 (0.9) ^a^	24.8 (1.7)	20.9 (1) ^c^
For healthy joints, arthritis	13.5 (0.7)	12.7 (0.9)	14.2 (0.7)	13.6 (0.9)	12.5 (1.4)	14.2 (1)
For healthy skin, hair, and nails	12.5 (0.7)	5.5 (0.5)	18.3 (1) ^b^	10.4 (0.7) ^a^	4.1 (0.6)	13.5 (1) ^c^
For bone health	22.8 (0.7)	13.6 (0.9)	30.5 (1) ^b^	35.4 (1.1) ^a^	20.5 (1.6)	42.9 (1.3) ^c^
To get more energy	23.2 (0.7)	21.3 (0.9)	24.8 (0.8) ^b^	16.5 (0.9) ^a^	14.3 (1.3)	17.7 (1.0) ^c^

Data are presented as percentages (SE). SE: standard error. *p*  <  0.05 was used to establish statistical significance. ^a^ indicates significant difference between self-initiated reasons for taking supplements and recommendations from doctors. ^b^ indicates significant difference between sexes within self-initiated decisions. ^c^ indicates significant difference between sexes within recommendations from doctors.

## Data Availability

The original data presented in the study are openly available in https://wwwn.cdc.gov/nchs/nhanes/Default.aspx.

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
