# Peer review of "Quantity, Duration, Adherence, and Reasons for Dietary Supplement Use among Adults: Results from NHANES 2011–2018"

_nutrients, 2024, doi:10.3390/nu16121830_

Round 1
Reviewer 1 Report
Comments and Suggestions for Authors
This study is very great. However, several changes are required prior the acceptance for publication.
-Abstract: is ok.
-Introduction: What is hypothesis of study?
-Methods: This topic is very small. To add the more details ref to methods used in NHANES.
-Results: To add the presence of disease and body composition of patients.
-Discussion: To add a discussion regarding to disease and body composition with use the supplements.
Author Response
-Introduction: What is hypothesis of study?
Our hypothesis was that demographic and socioeconomic characteristics can impact quantity, duration, adherence, and the reasons for taking dietary supplements (Lines 68-70).
-Methods: This topic is very small. To add the more details ref to methods used in NHANES.
Thank you for your comments. We have added more detailed information on the methods used in NHANES (Line 77-84; Lines 127-136; Lines 139-141).
-Results: To add the presence of disease and body composition of patients.
Thank you for your constructive comments. We appreciate the relevance of these metrics in understanding the overall health context of the study population. However, we would like to clarify that the primary aim of our study was to explore patterns of dietary supplement use among adults and the impact of demographic and socioeconomic characteristics on supplement use, rather than to investigate the direct implications of disease and body composition on supplement use. Given the scope of our study, we have acknowledged this as a limitation in our discussion section. We believe that this acknowledgment appropriately addresses the potential concerns and suggests a need for future research to explore these aspects (Lines 421-424).
-Discussion: To add a discussion regarding to disease and body composition with use the supplements.
Thank you for your suggestion. However, as mentioned before, the primary focus of our research was to examine the patterns and reasons behind dietary supplement use, rather than exploring the impacts of disease and body composition on supplement use. Although we agree that BMI and the presence of certain health conditions are important factors that could influence dietary supplement consumption, these were not within the specific objectives of our current study. Considering your comments, we have mentioned this in the limitations section of our manuscript, suggesting that future research could specifically address how disease and body composition may interact with dietary supplement use (Lines 421-424).
Reviewer 2 Report
Comments and Suggestions for Authors
The study "Quantity, duration, adherence, and reasons for dietary supplement use among adults: Results from NHANES 2011-2018” not only provides a robust and comprehensive foundation for the analysis but also offers novel insights into dietary supplement usage. Its detailed examination across various demographic and socioeconomic segments significantly enriches the understanding of the subject. The manuscript is well-organized and clearly written, with complex analyses and a multitude of data. The study also addresses an essential aspect of public health by highlighting the proactive use of dietary supplements for self-managed health and the need for healthcare professionals to guide such behaviours.
However, a few points could be critically addressed or improved:
1. While the manuscript mentions using univariate and multivariate logistic regression models to explore associations, a more detailed discussion of the choice of variables included in these models would be beneficial. Clarification on whether stepwise regression, fixed entry, or another method was used to select variables for the final models would also be beneficial.
2. The treatment of missing data is not explicitly discussed. In large datasets like NHANES, how missing data are handled (e.g., imputation techniques, complete case analysis) can significantly affect the analysis results. Addressing this would strengthen the study's methodological transparency and reliability.
3. While odds ratios are provided, little is discussed regarding the clinical significance of these findings. Including effect sizes or discussing the practical implications of the observed associations in a healthcare context would make the results more applicable.
4. The study, while extensive, does not explore the long-term health outcomes of dietary supplement use.
Author Response
- While the manuscript mentions using univariate and multivariate logistic regression models to explore associations, a more detailed discussion of the choice of variables included in these models would be beneficial. Clarification on whether stepwise regression, fixed entry, or another method was used to select variables for the final models would also be beneficial.
We appreciate your positive remarks about our study and manuscript. Thank you for your comment on the statistical approaches utilized in our manuscript. Initially, we conducted univariate logistic regression analyses to assess the individual effects of each variable on the reasons for dietary supplement intake. For the multivariate regression analysis, we included all five variables, adjusting each one independently in the model. This approach allowed us to obtain adjusted odds ratios that reflect the influence of each variable while controlling for the effects of the others. We did not use stepwise regression; instead, we utilized a fixed entry method for variable inclusion (Lines 127-132).
- The treatment of missing data is not explicitly discussed. In large datasets like NHANES, how missing data are handled (e.g., imputation techniques, complete case analysis) can significantly affect the analysis results. Addressing this would strengthen the study's methodological transparency and reliability.
Thank you for your comments. After a preliminary assessment of the dataset, we observed that the incidence of missing values was relatively low. Consequently, we opted for a complete case analysis approach (Lines 139-141).
- While odds ratios are provided, little is discussed regarding the clinical significance of these findings. Including effect sizes or discussing the practical implications of the observed associations in a healthcare context would make the results more applicable.
Thank you for highlighting the need to discuss the clinical significance of our findings. We have added the clinical significance of these findings in the discussion (Lines 361-368).
- The study, while extensive, does not explore the long-term health outcomes of dietary supplement use.
We acknowledge the limitation regarding the exploration of long-term health outcomes of dietary supplement use, as noted in our study. We have discussed this in the limitations section of our manuscript. Future studies could potentially address this gap to provide a more comprehensive understanding of the long-term implications of dietary supplement consumption (Lines 426-427).

Reviewer 3 Report
Comments and Suggestions for Authors
The authors found an interesting research idea and completed analysis of NHANES data to answer a discrete question about dietary supplement use. The article is well written overall, but there are areas of improvement.
Line 121. Reported 61.9% of individuals "held a college degree or higher" is inconsistent with the data in Table 1. This summary combines data of those that just attended college with those that have a college degree. Only 29.7% of subjects held a college degree.
Figure 2. It says days is the group variable in the Figure legend, but these graphs would be much easier to read if it said the units (days) after each number in each figure key.
Figure 3. Same note about including days in the key for each figure.
Figure 4B. There are extraneous red dots in many of the figures. I thought it was an in image rendering issue, but it seems intentional as this figure has a large, hand-drawn white glob and a red line randomly placed in the figure.
Figure 4B. Should also list years old in the key for this figure.
ALL FIGURES (Line 223). Need to specify what the error bars in the figure represent in the legend of each figure. Each figure should be able to stand alone. As it is currently presented, readers have pass 8 pages of figures before an explanation of the error bars is presented.
Lines 224-227. It is essential that this information be included in the legend for Figure 4 and not in the text of the document after all of the images for Figure 4.
Line 230. This may have been mentioned before, but it should be reiterated again in this section. Could people choose more than one reason?
Table 2. The column headers are too succinct. Authors should consider making these columns headers 2 lines. "Prevent" and "Boost" are particularly too generic. Disease Prevention and Boosting Immunity is essential to include for this table to stand on its own.
Line 243. It is essential for this notation to be included in the Table legend.
Lines 244-247. This description also belongs in the legend and not in the text of the document.
Line 273. Why were significant differences not reported on any of the other figures?
Lines 375-382. All of these statements are relevant, but they seem out of context here. This paragraph does not seem to fit with the flow of the discussion of your study results.
Lines 384-392. This paragraph also reads as more of a lit review and does not seem to fit within the context of the discussion in this paper.
Line 396. "However, discussions about dietary supplements only occurred in..." This is speculation. These discussions were not observed by the researcher. It is more accurate to state that only 25% of survey respondents reported the interactions.
Line 412. Need more practical applications in the conclusion. Who is the audience? What was the purpose of asking these research questions? Are the data meaningful? To whom are the data meaningful?
Supplemental Table 2. Define OR in the table legend
Supp. Table 7. Must be amended to fit clearly on a page.
Comments on the Quality of English LanguageThe article was well written for the most part. There are a few instances of errors with subject-verb agreement throughout the article that should be addressed (e.g. Line 96, 132, etc.).
Line 134. "were with" is also awkwardly phrased.
Author Response
Line 121. Reported 61.9% of individuals "held a college degree or higher" is inconsistent with the data in Table 1. This summary combines data of those that just attended college with those that have a college degree. Only 29.7% of subjects held a college degree.
Thank you for bringing this discrepancy to our attention. Upon reviewing the manuscript and associated data, we acknowledge an error in Line 121. The reported 61.9% incorrectly combined the data for individuals who have some college with those who have obtained a college or higher degree (Line 146)
Figure 2. It says days is the group variable in the Figure legend, but these graphs would be much easier to read if it said the units (days) after each number in each figure key (Line 169).
Thank you for your suggestion regarding the clarity of Figure 2. We have updated the figure to include the unit "days" next to each numerical value in the Figure key (Line 169).
Figure 3. Same note about including days in the key for each figure.
Thank you for your suggestion. We have updated the figure to include the unit "days" next to each value in the figure key (Line 174).
Figure 4B. There are extraneous red dots in many of the figures. I thought it was an in-image rendering issue, but it seems intentional as this figure has a large, hand-drawn white glob and a red line randomly placed in the figure.
Thank you for pointing out the visual concerns in Figure 4B. Upon review, the red dots and the hand-drawn elements you mentioned were not intended to be part of the final figure. These markings resulted from an oversight in our figure preparation process. We have revised Figure 4B, removing all extraneous marks and ensuring the figure accurately represents the data without any unintended distractions. We have also reviewed other figures to prevent similar issues. We apologize for any confusion this may have caused and appreciate your attention to detail in helping us maintain the quality of our visual data presentation.
Figure 4B. Should also list years old in the key for this figure.
Thank you for your suggestion. We have updated the figure (Line 221).
ALL FIGURES (Line 223). Need to specify what the error bars in the figure represent in the legend of each figure. Each figure should be able to stand alone. As it is currently presented, readers have pass 8 pages of figures before an explanation of the error bars is presented.
Thank you for your insightful feedback regarding the clarity of our figures. We agree that each figure should be self-explanatory to ensure that readers can fully understand the data presented without the need to refer to other sections of the manuscript for basic explanations. Thus, we have updated the legends of all figures to clearly specify that the error bars represent standard error.
Lines 224-227. It is essential that this information be included in the legend for Figure 4 and not in the text of the document after all of the images for Figure 4.
Thank you for your comments. It has been revised as suggested (Line 213).
Line 230. This may have been mentioned before, but it should be reiterated again in this section. Could people choose more than one reason?
Yes. participants could choose more than one reason (Line 199).
Table 2. The column headers are too succinct. Authors should consider making these columns headers 2 lines. "Prevent" and "Boost" are particularly too generic. Disease Prevention and Boosting Immunity is essential to include for this table to stand on its own.
Thank you for your comments. We have revised it as suggested.
Line 243. It is essential for this notation to be included in the Table legend.
Thank you for your suggestions. It has been addressed (Line 207).
Lines 244-247. This description also belongs in the legend and not in the text of the document.
We have modified the manuscript as suggested.
Line 273. Why were significant differences not reported on any of the other figures?
Thank you for your comments. To address the issue of not reporting significant differences in other figures, we have added Table 2 in the manuscript. This new table provides a comparative analysis of the quality of supplements, duration of use, and the number of days any supplements were taken in the past 30 days. We opted to present the data using medians and interquartile ranges (IQR) in Table 2, as these measures offered a more accurate reflection of the central tendency and variability within our non-normally distributed data, compared to using categorical variables. This approach enhances the clarity and precision of our data presentation, ensuring that our statistical analyses accurately convey the trends observed in our study (Line 179).
Lines 375-382. All of these statements are relevant, but they seem out of context here. This paragraph does not seem to fit with the flow of the discussion of your study results. Lines 384-392. This paragraph also reads as more of a lit review and does not seem to fit within the context of the discussion in this paper.
Thank you for your constructive feedback on the organization and relevance of the content in the discussion section of our manuscript. Upon review, we agree that the paragraphs in lines 375-382 and 384-392 did not seamlessly integrate with the overarching narrative of our study's Results and Discussion sections. As you suggested, we have removed these paragraphs from the Discussion section.
Line 396. "However, discussions about dietary supplements only occurred in..." This is speculation. These discussions were not observed by the researcher. It is more accurate to state that only 25% of survey respondents reported the interactions.
Thank you for your suggestion. To address this, we have revised line 397 to reflect the data more accurately: "Only 25% of survey respondents reported having interactions about dietary supplements.”
Line 412. Need more practical applications in the conclusion. Who is the audience? What was the purpose of asking these research questions? Are the data meaningful? To whom are the data meaningful?
Thank you for your comments. This study provides valuable insights into the real-world patterns of supplement use, highlighting the necessity for healthcare providers to discuss supplement use during routine consultations and offer evidence-based guidance. By understanding these patterns, healthcare providers can deliver a tailored advice, and public health officials and policy makers can utilize these insights to formulate public health messages and regulatory actions that ensure the safety and efficacy of dietary supplements on the market (Lines 404-414).
Supplemental Table 2. Define OR in the table legend.
We have added a definition for "OR" (Odds Ratio) in the table legend to ensure that all readers understand the statistical measure used in our analysis.
Supp. Table 7. Must be amended to fit clearly on a page.
We have reformatted this table to ensure that it fits clearly on a page.
Comments on the Quality of English Language
The article was well written for the most part. There are a few instances of errors with subject-verb agreement throughout the article that should be addressed (e.g. Line 96, 132, etc.). Line 134. "were with" is also awkwardly phrased.
Thank you for your feedback on the language quality. We appreciate your pointing out the specific issues with subject-verb agreement and other grammatical concerns. We have carefully reviewed the manuscript and corrected these errors. For example, in line 157, "were with" has been revised to "had" to enhance readability.
